# Tourists and Local Stakeholders' Perception of Ecosystem Services Provided by Summer Farms in the Eastern Italian Alps

**Carine Pachoud** [1] , **Riccardo Da Re** [2], **Maurizio Ramanzin** [1], **Stefano Bovolenta** [3] , **Damiano Gianelle** [4] **and Enrico Sturaro** [1,*]

1   Department of Agronomy, Food, Natural Resources, Animals and Environment, University of Padova, Viale dell'Università, 16, 35020 Legnaro (PD), Italy; carine.pachoud@hotmail.fr (C.P.); maurizio.ramanzin@unipd.it (M.R.)

2   Department of Land, Environment, Agriculture and Forestry, University of Padova, Viale dell'Università, 16, 35020 Legnaro (PD), Italy; riccardo.dare@unipd.it

3   Department of Agricultural, Food, Environmental and Animal Sciences, University of Udine, Via delle Scienze, 206, 33100 Udine, Italy; stefano.bovolenta@uniud.it

4   Department of Sustainable Agro-Ecosystems and Bioresources, Research and Innovation Centre, Fondazione Edmund Mach, 38010 San Michele all'Adige (TN), Italy; damiano.gianelle@fmach.it

*   Correspondence: enrico.sturaro@unipd.it; Tel.: +39-049-8272641

**Abstract:** In the Alps, summer farms are temporary units, where cattle are moved during summer to graze on Alpine pastures, which provide multiple ecosystem services (ESs), many of which do not have a market value. This study aimed at understanding and comparing the perceptions of summer farms and of the associated ESs by local stakeholders and tourists in a study area of the province of Trento in the eastern Italian Alps. Thirty-five online questionnaires and two focus groups were realized with local stakeholders involved in the dairy value-chain. Semi-structured interviews were conducted with 405 tourists in two representative summer farms. The perceptions of summer farms differed between local stakeholders, who mainly focused on provisioning ESs, and tourists, who mainly focused on cultural and regulating ESs. Both categories of actors rated positively eight different ESs associated with summer farms, but demonstrated a lack of knowledge of specific regulating ESs. This study showed that discussion among the different actors is required to increase mutual knowledge and to grasp the diversity of links between summer farms and ESs, in order to support public policies and private initiatives for promoting summer farm products and the sustainable development of mountain regions.

**Keywords:** ecosystem services; summer farm; Alpine pasture; local stakeholders; tourists

## 1. Introduction

Livestock systems based on the management of semi-natural grasslands are multifunctional and provide important ecosystem services (ESs) to the society [1–5]. Ecosystem services are defined as the direct and indirect contributions of ecosystems to human well-being, and are classified into four groups [6]: Supporting ESs (e.g., soil formation, photosynthesis and nutrient cycling), provisioning ESs (e.g., food, water, timber, fiber), regulating ESs (e.g., climate regulation, flood prevention and water purification) and cultural ESs (e.g., recreational, cultural and spiritual benefits). The regulating, cultural and supporting ESs do not have a market value and are therefore often ignored within evaluation frameworks [4,7].

In the Italian Alps, as in most mountain regions of Europe, dairy cattle are traditionally moved seasonally to summer farms, which are temporary units, located mostly between 1200 and 2000 m of altitude, that rely on the grazing of Alpine pastures to feed the animals, usually from June to September [8]. These pasture-based livestock systems have contributed to the establishment and maintenance of semi-natural grasslands rich with environmental and cultural values, and provide multiple ESs [3,5], including provisioning ESs, e.g., the production of milk and traditional cheese [1–3], regulating and supporting ESs, e.g., soil carbon sequestration [9], protection from landslides and fires [10,11]), conservation of natural habitats and biodiversity [9,12–14], and also cultural ESs, e.g., aesthetic value [15] and touristic attractiveness [16] of the landscape, conservation of the cultural heritage [17,18]. However, during the last decades traditional livestock farming systems have experienced a strong decline, with the consequent abandoning of Alpine grasslands [19–22]. In the Italian Alps, for instance, 27% of meadows and pastures were abandoned between 1990 and 2010 [1]. This abandoning often results in a loss of the ESs associated with the traditional farming systems and their management of semi-natural grasslands [23–25].

Today, the application of the ESs framework to research on mountain regions and environments is of growing interest [26,27]. Mengist et al. [28] found that 1252 papers linked to mountain ESs were published between 1992 and June 2019, and Martín-López et al. [29] that 993 papers were published between 1997 and 2018, 42% of which in 2017 and 2018. In this respect, understanding and valuing the multiple ESs linked with traditional farming practices is a necessary step for the conservation of the multiplicity of public and private benefits of mountain regions. The ESs framework is instrumental in understanding and supporting the economic, ecologic and social sustainability of such farming systems, and of the habitats that they conserve. In particular, remunerating the non-marketable ESs provided by extensive livestock systems is necessary for their economic sustainability, and hence for ensuring the future provision of the desired ESs. Since the benefits of these ESs are public, their remuneration should come first from the implementation of payments for ESs in policy measures [2,30,31]. For example, the European agricultural fund for rural development (EAFRD) and the rural development program (RDP), as implemented by the Autonomous Province of Trento, provide payments for the conservation and improvement of meadows and pastures. These payments, which are presently of 90€/ha of grazed Alpine pasture for a herd with a minimum of 15 lactating cows and 75€/ha for a herd of less than 15 lactating cows [32], could be differentiated and improved by linking them to the provision of a set of desired ESs. In addition to coming from policy measures, remuneration of non-marketable ESs could also come from private marketing strategies promoting a higher price of a food product on the basis of the ESs associated with its specific productive chain. A common requisite for the success of public and market actions promoting the remuneration of ESs is that the society is willing to pay for the targeted ESs [26,27]. In this respect, studies conducted in the province of Trento as well as in other European mountain areas [30,31] showed that the willingness of consumers to pay for various regulating and supporting ESs was high, although the perception of the specific ESs differed regionally. In the context of the province of Trento, the consumers' total willingness to pay for a set of four ESs was close to 160€.

In the process of economic valuation of the non-marketable ESs, a preliminary step is the knowledge of the perception of specific ESs and of the social value attributed to them by the different social actors. In particular for summer farms, this requires considering both the local stakeholders involved in the dairy value chain and the tourists who visit summer farms [33]. The aim of this study was to analyze and compare the perceptions of local actors and tourists of the summer farms and the associated ecosystem services in an Alpine area where both dairy farming and tourism are important for the valorization of local economy, landscape and biodiversity. Participatory approaches (focus groups) and online questionnaires were conducted with local stakeholders involved in the value chain, and self-filled questionnaires with tourists visiting summer farms.

## 2. Materials and Methods

### 2.1. Study Area

The Autonomous Province of Trento is located in the Alps of north-eastern Italy (Figure 1). It covers an area of 6.200 km$^2$ and comprises 217 municipalities. Utilized agricultural areas cover 1400 km$^2$, predominantly characterized by meadows and pastures (81%), followed by orchards and vineyards (17%), and arable crops (2%) [34]. According to the Breeders Federation of Trento [35], dairy cattle breeding is the main component of the livestock sector in the province, with 800 dairy farms out of a total of 1.400 farms. Dairy farming is strongly associated with dairy cooperatives, which collect 80% of the total milk produced in the province and process it into typical cheeses, many of which are PDO labelled [36].

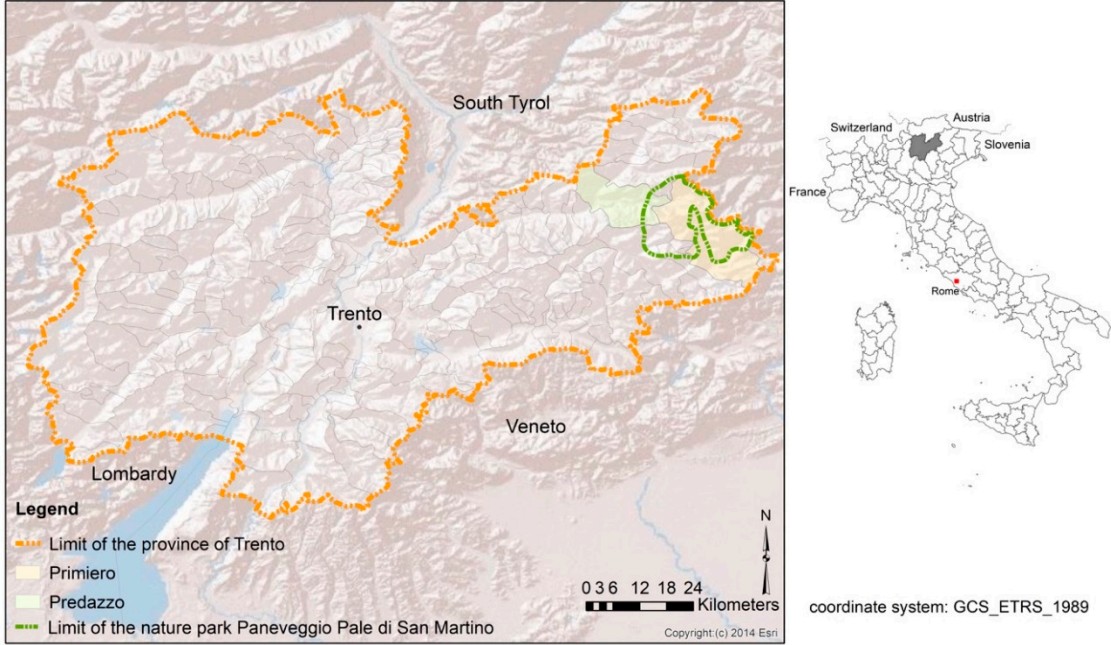

**Figure 1.** Localization of the province of Trento, the natural park Paneveggio Pale di San Martino and the municipalities of Primiero and Predazzo (source: Own elaboration).

In the province of Trento, 305 summer farms are currently active and play an essential role for the local dairy farming sector [37]. Only half of them are suitable for lactating cows, and host about 11.500 heads (47% of the total). The other half host dry cows or replacement cattle [35]. In 2018, the milk produced in summer farms was around 9.000 T (6% of the yearly total production in the province). Processing of milk into cheese and butter is carried out in about 90 summer farms, but only on one or two days per week. Most of the milk produced in summer farms (80%) is collected and processed by the cooperative dairies. Some of them process it separately from that collected in lowland permanent farms. The cheeses and dairy products obtained with summer farm milk are a niche production (about 4% of total dairy products) that is marketed under the brand "Sapori di Malga" ("flavors of the summer farm") and is expected to increase in the next years [36].

Around 30 summer farms practice agritourism activities (i.e., accommodation, restaurant and degustation) [38]. The province of Trento has established the brand "Agritur Trentino" that is a tourism offer of rural structures where tourists may taste local products and watch, or even participate in, their processing.

Within this context, the study was specifically conducted in two case studies, which are the municipalities of Primiero and Predazzo (Figure 1). They are representative of the provincial context where dairy production is an important economic activity, strongly connected with milk processing into typical cheeses in dairy cooperatives [39], and the practice of moving cattle to summer farms is widespread [37]. In both municipalities, the local dairy farmers are associated to the cooperative

dairies "Caseificio Sociale di Primiero" and "Caseificio Sociale di Predazzo e Moena" that produce typical cheeses with summer farm milk. In these municipalities, sixteen summer farms presently keep dairy cattle, and almost all of them offer agritourism activities. In both municipalities, in addition, summer tourism is an important economic activity.

### 2.1.1. Online Questionnaire with the Local Stakeholders

A group of 67 local stakeholders were identified with a judgment sampling (nonprobability sampling) approach, carried out by the natural park Paneveggio Pale San Martino, which was responsible for providing a first set o stakeholders involved (directly or indirectly) in the dairy chains and in the summer farms management. A name grid generator matrix divided into categories was used to facilitate the creation of a complete list of stakeholders, who were selected in a second step with a simple "insider-outsider matrix" [40]. Preliminary meetings were organized with the partners involved in the project (natural park Paneveggio Pale San Martino, Breeder Federation of Trento, Edmund Mach Foundation, University of Padova and University of Udine) to identify the relevant stakeholders. Stakeholders were divided into professional categories (farmers, summer farm managers, dairy cooperative representatives, local authority representatives, tourism and cultural operators) and were chosen considering their different involvement in the value chain and their proximity to the issue stakes.

The identified stakeholders were asked by email to answer to an online questionnaire aimed at analyzing their level of knowledge and their perception of summer farms and of the associated ESs. We chose to use online questionnaires because this method has various advantages (e.g., low cost, speed of data collection) [41]. We tried to overcome its main limitations (e.g., low motivation of respondents, internet access and capacity to reach all selected stakeholders) through personalized emails signed by the Park director. The questionnaire was divided into three parts. In the first, the respondents were asked to give three words (hereafter called "induced words") related to summer farms. In the second, they were asked to answer to the question "can you give the definition of ES?", choosing one among the answers "Yes"; "I know the meaning roughly"; "I've heard of this concept, but I don't know its meaning"; "I never heard about it". Then, they could read the definition of ESs according to TEEB [7]. In the third, they were asked to express their level of agreement, on a five-point Likert scale, with a list of positive effects of the summer farms on eight ESs. The score varied from −2: Very negative to +2: Very positive. The option "I don't know" was included. The ESs were chosen from a previous review of literature gathering provisioning, regulating, supporting and cultural services [4,6,26,42], and are listed in Table 1.

**Table 1.** List of the eight Ecosystem services (ESs) chosen for the assessment by stakeholders.

| ESs Type | | Specific ESs |
|---|---|---|
| Provisioning ESs | 1. | Production of high-quality dairy products |
| Supporting ESs | 2. | Maintaining a high biodiversity through the preservation of habitats for wildlife (plants and animals) |
| Cultural ESs | 3. | Maintenance of traditional landscape |
| | 4. | Increase in tourism activities |
| | 5. | Respect for animal welfare |
| Regulating ESs | 6. | Control of shrub invasion on pastures |
| | 7. | Improved soil carbon storage |
| | 8. | Contribution to mitigation of climate change |

A total of 35 questionnaires were completed. The categories of stakeholders, according to their profession, are listed in Table 2.

**Table 2.** Category of stakeholders who answered to the online questionnaire and were involved in focus groups.

| | Questionnaire | Focus Groups | |
| --- | --- | --- | --- |
| **Stakeholders** | **Number of Respondents** | **Primiero** | **Predazzo** |
| Farmers | 4 | 1 | 0 |
| Summer farm managers | 2 | 2 | 0 |
| Dairy cooperative representatives | 3 | 1 | 1 |
| Local authority representatives [1] | 5 | 3 | 5 |
| Tourism operators | 13 | 2 | 4 |
| Cultural operators | 7 | 2 | 1 |
| Others [2] | 1 | 0 | 0 |
| Total | 35 | 11 | 11 |

[1] Policy makers: members of municipality council, agent of the Autonomous Province of Trento. [2] The answer was not specified.

### 2.1.2. Focus Groups with the Local Stakeholders

Based on the results of the online questionnaire, and addressed to the same target groups of stakeholders, two focus groups were conducted in March 2019, one in Primiero and one in Predazzo. The sample for the focus groups was based on voluntary participation. The focus groups were held with 11 local stakeholders in both Primiero and Predazzo (Table 2). However, none of the farmers and summer farm managers volunteered to participate in the Predazzo focus group. A focus group is an in-depth interview of a small group of people (4–12) [43] that aims at analyzing the opinion of the group and not those of the individuals. The idea is to keep the group small enough to allow everyone to talk, but big enough to capture the range of opinions [44]. The objective of the focus groups was to obtain a deeper understanding of the perceptions by local stakeholders of the ESs provided by the summer farms.

A common design for the two focus groups was created (Table 3). The groups were managed by a moderator and an assistant who observed, and took notes on, the dynamics that emerged during the discussion among the participants. Each focus group began with a short introduction to present the project, give the definition of ESs [7], and summarize the results of the online questionnaire.

**Table 3.** Design of the focus groups held with local stakeholders.

| Session | Comments | Leading Questions |
| --- | --- | --- |
| Session 1: Important Ecosystem Services (ESs) provided by alpine pastures (30 min) | Discussion using post-it: Each participant received 3 post-it cards; the participants noted short statements and presented their comments; the facilitators grouped the cards by resemblance on a flip chart. | What are according to you the main ESs provided by the Alpine pastures, that could be used to improve communication on summer farm products? |
| Session 2: Assessment of ESs (30 min) | Online survey results of the assessment were presented on a poster. The group was requested to provide a new ranking if wanted. | Do you agree with the current ranking? If not, please provide a new ranking. |

### 2.1.3. Self-Filled Questionnaires with the Tourists

A total of 405 questionnaires with Italian tourists aged over 18 years were conducted. We wanted to assess the perception of tourists visiting summer farms and for this purpose we chose two summer farms, Juribello and Vallazza, which are representative of the context, since they keep lactating cows and are regularly visited by tourist in summer, have a farmhouse restaurant, sell cheese and offer once per week a demonstration of cheese making. Questionnaires were self-filled by tourists during

lunch, in the presence of the interviewer, on twenty consecutive days in July 2019. This period of random collection, and the number of questionnaires which is larger or similar to that reported in other studies [45,46], were chosen to ensure that the sample was representative.

The questionnaire was structured in four parts. In the first, the aim of the study and the general definition of ESs according to TEEB [7] were proposed. In the second, the tourists were asked to give three induced words related to summer farms. In the third, they were asked to express their agreement on the positive effects of the summer farms on the eight ESs listed in Table 1, with the same Likert scale. Lastly, the fourth part recorded gender, age, province of residence and level of study of the respondents to stratify the sample for statistical analysis.

### 2.1.4. Comparison between the Perceptions of the Local Stakeholders and Tourists

The induced words given by local stakeholders and tourists were gathered into semantic groups. Since the multitude of induced words cannot be submitted to statistical analysis, the words having the same meaning or being semantically close are gathered under a common term, called semantic group [47]. The frequencies of the identified semantic groups were compared between stakeholders and tourists using a Chi-squared test. We considered as expected the frequencies of semantic groups of stakeholders, who had a sample size much smaller than that of tourists. The scores assigned to the effects of summer farms on the eight ESs by stakeholders and tourists were compared with the non-parametric test of Kruskal–Wallis, computed by PROC UNIVARIATE (SAS 9.2; SAS Institute Inc., Cary, NC, USA).

## 3. Results

### 3.1. Questionnaires with Local Stakeholders and Tourists

### 3.1.1. Perceptions of Summer Farms

We obtained a total of 90 words and 15 blanks from the 35 online questionnaires filled by local stakeholders, and a total of 1171 words and 44 blanks from the 405 questionnaires self-filled by tourists. The tourists came from 46 different provinces (11 provinces in the Alps); 17.2% of them came from the Trento province. Genders were equally represented (49.1% female and 50.9% male). Mean age was 50.6 years (minimum: 18; maximum: 86). Level of study was 51% high school, 33% graduation, 14% secondary school, 1% PhD, 1% primary school, 1% other. From the induced words we formed 14 semantic groups, which are listed in Table 4 together with the most frequent induced words. The semantic groups "collective property" (frequencies: 0.01 in local stakeholders and 0.00 in tourists), "animal welfare" (frequencies: 0.01 in both local stakeholders and tourists) and "family" (frequencies: 0.01 in both local stakeholders and tourists) had a frequency < 0.05. We removed them from the Chi-squared comparison between stakeholders and tourists in order to obtain more robust results. In addition, we preliminarily verified with Chi-Squared tests that the frequencies of semantic groups within the tourist's sample did not differ between genders, province of residence and levels of study. We found that the perceptions of summer farms depended strongly on the category of actors ($\chi^2 = 1320.7$, N = 9, $p < 0.001$).

**Table 4.** Semantic groups and most frequent induced words describing a summer farm obtained from the questionnaires filled by local stakeholders and tourists.

| Semantic Groups | Most Frequent Induced Words | |
| --- | --- | --- |
| | Local Stakeholders | Tourists |
| Alpine pastures | Alpine pastures | Alpine pastures |
| Products | Products, cheese, production, milk | Dairy product, local products, quality products, km0 products, good food, milk, typical products, cheese, salami, typical dishes, production |
| Livestock | Livestock, animals, herd, husbandry, cows | Animals, livestock, husbandry, cows, grazing animals |

**Table 4.** *Cont.*

| Semantic Groups | Most Frequent Induced Words | |
| --- | --- | --- |
| | **Local Stakeholders** | **Tourists** |
| Building | Building, structure, barn | Typical house, building, structure, barn |
| Environment | Nature, territory, environment, altitude | Nature, altitude, landscape, scenery, non-polluted environment, mountain, pure air, green, Dolomites |
| People involved in the value chain | Farmers, shepherds | |
| Sustainability | Management of the territory, care of the territory | Maintenance, preservation, ecosystem protection, services, useful, eco-sustainability, self-sufficiency, small-scale farming, natural production, offer of jobs, respect of the environment, maintenance of human life in the mountains |
| Culture | Culture, tradition | Tradition, way of life, simple life, culture, valorization of the traditions, history, memory, folklore |
| Tourism | Gastronomy, restaurant | Hosting, hospitality, for children, restaurant, clean, meeting point, hiking, reference point, comfortable, fountain, availability |
| Authenticity | Authenticity | Authenticity |
| Animal welfare | Animal welfare | Animal welfare, happy animals |
| Family | Family activity | Family activity |
| Collective property | Collective property | |
| Tranquility | | Peace, relax, tranquility, silence, well-being, safety |

The average frequencies of semantic groups expressed by local stakeholders and tourists are shown in Figure 2. The semantic groups "Alpine pastures", "products" and "livestock" were the most frequently used to describe a summer farm by the local stakeholder, while the groups "products", "environment" and "tourism" were the most frequently used by tourists. Therefore, stakeholders have a representation of summer farms based on the related provisioning ESs, while tourists have a representation more related to cultural ESs.

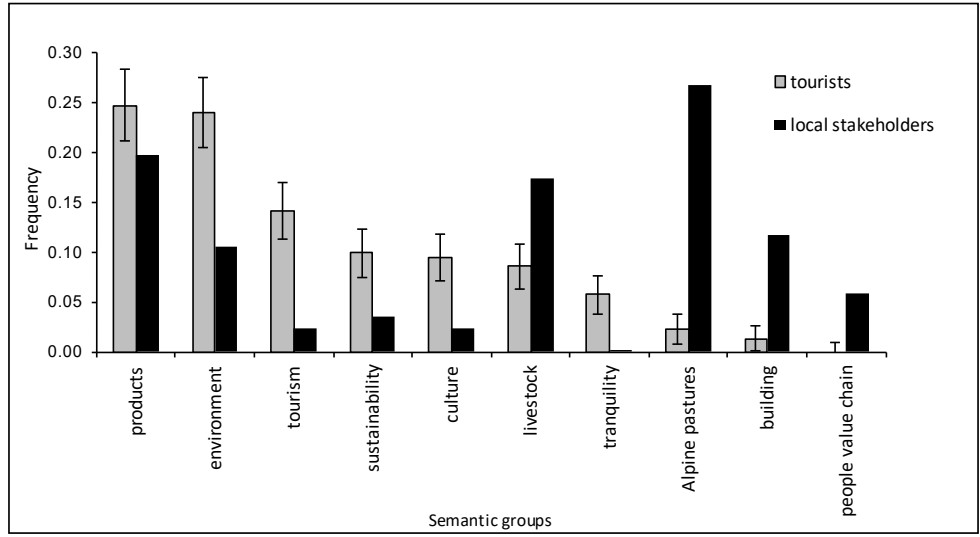

**Figure 2.** Average frequencies of semantic groups related to summer farms expressed by local stakeholders and tourists. The frequencies of stakeholders were taken as reference in a Chi-squared test, and the error bars of the frequencies of tourists indicate the confidence intervals (95%).

### 3.1.2. Perception of ESs Provided by Alpine Pastures

Among the thirty-five local stakeholders that filled the online questionnaire, eleven declared to know roughly the meaning of the ESs concept, twelve to have heard of it, but to have no knowledge

of its meaning, and three to have never heard of it. Only nine declared to know the definition of ESs (Figure 3).

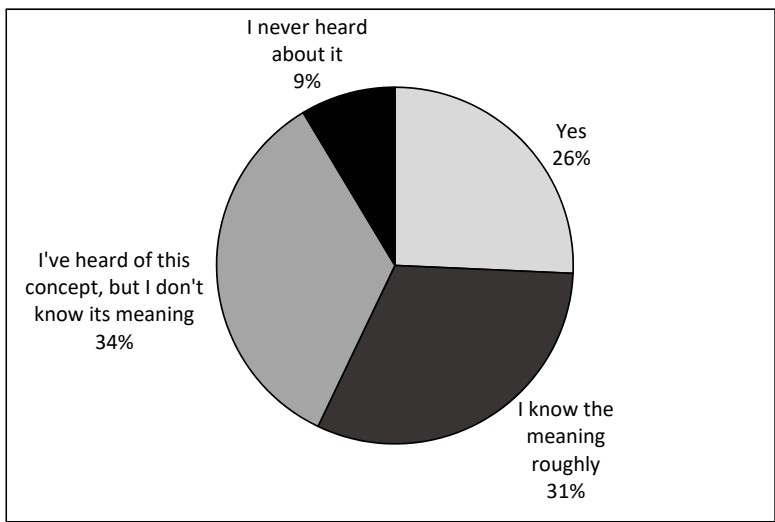

**Figure 3.** Level of knowledge on Ecosystem Services obtained from the online questionnaire filled by local stakeholders.

The proportions of local stakeholders choosing the answer "I don't know" when asked to express their level of agreement on a proposed ES related to summer farms are shown in Figure 4. The proportions of respondents declaring lack of knowledge were remarkable for the regulating ESs "improved soil carbon storage" and "contribution to mitigation of climate change" (Figure 4), especially in tourists for the question on carbon storage (59% of "I don't know" answers). In addition, while 10% and 9% of tourists declared lack of knowledge on the ESs "control of the shrub invasion on pastures" and "maintaining a high biodiversity through preservation of habitats for wildlife (plants and animals)", respectively, all the stakeholders were able to assess both ESs.

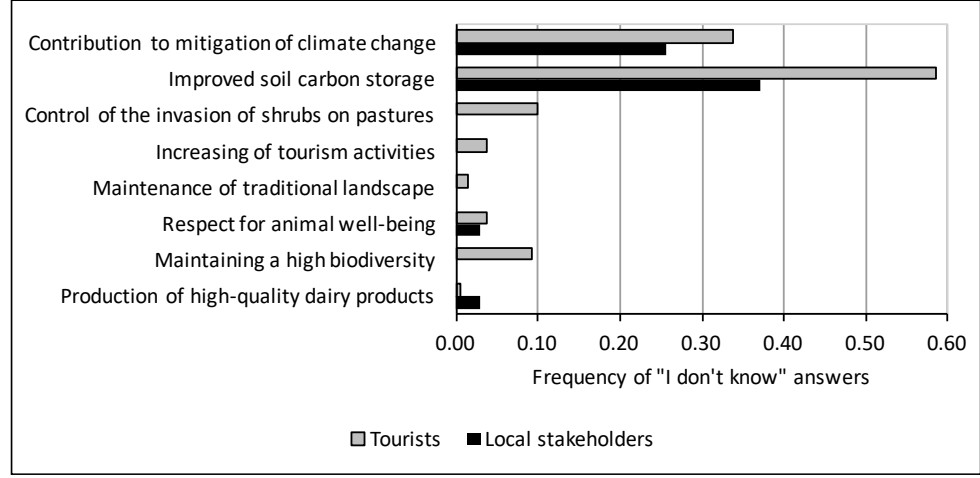

**Figure 4.** Number of "I don't know" answers given by local stakeholders and tourists when asked to assess the eight ESs proposed in association with summer farms.

When evaluated, all the proposed ESs were positively assessed by both local stakeholders and tourists, since the scores given were always positive (Figure 5). A tendency for less positive scores was apparent for the less known ESs, e.g., "contribution to mitigation of climate change" and "improved soil carbon storage". The Kruskal–Wallis test indicated that "the quality of dairy products" and "Maintaining a high biodiversity" were significantly more important for tourists, whereas "control of

the shrub invasion on pastures" was more important for local stakeholders. The differences between the scores given by local stakeholders and tourists to "contribution to mitigation of climate change" and "improved soil carbon storage", although remarkable, did not reach the statistical significance. This is most probably because for these two ESs the sample size was greatly reduced, since the many respondents that declared their lack of knowledge (see Figure 4) did not provide a score.

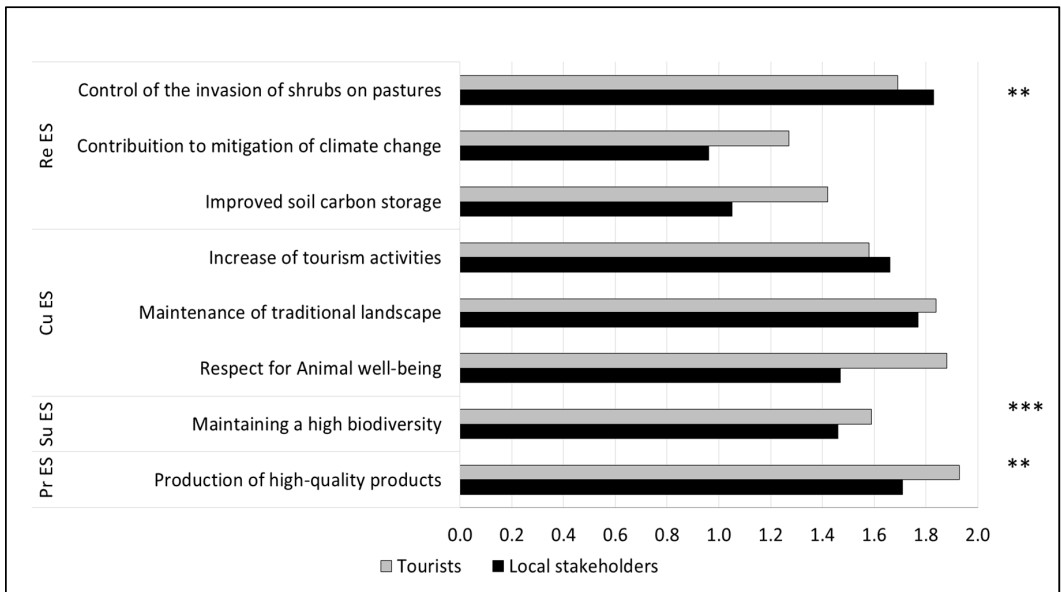

**Figure 5.** Assessment of the eight ecosystem services (ESs) by the local stakeholders and the tourists. Re ES: Regulating ES; Cu ES: Cultural ES; Su ES: Supporting ES; Pr ES: Provisioning ES. The bars indicate the average scores on a five-point Likert scale (from -2 to +2). *** and ** indicate that the difference between local stakeholders and tourists is significant at $p < 0.001$ and 0.01, respectively.

### 3.2. Focus Groups with the Local Stakeholders

During the first session, we observed that both the focus groups had similar perceptions, linked to the environmental and economic sustainability, the intrinsic quality of the products, the identity and culture of the people working on summer farms and the animal welfare (Table 5).

**Table 5.** Ecosystem services provided by summer farms as indicated by the focus groups with local stakeholders held in Primiero and in Predazzo.

| Primiero | Predazzo |
| --- | --- |
| 1. environmental sustainability: the use of pastures allows the maintenance of the landscape; | 1. environmental sustainability: The use of pastures allows the maintenance of the territory with the preservation of biodiversity, water and landscape diversity; |
| 2. economic sustainability through the maintenance of human presence in the mountains and the creation of a local economy reinforced by tourism attractiveness; | 2. economic sustainability: The production of summer farm products allows the maintenance of human presence in the mountains and gives job opportunities to a skilled workforce and to young generations. Moreover, the tourism attractiveness allows to promote the quality and the typicality of products. It allows also to reach a niche market creating added value; |
| 3. intrinsic quality of the product (organoleptic, healthiness) thanks to the use of Alpine pastures; | 3. animal welfare; |
| 4. cultural aspects and traditions that give a specific identity to the people; | 4. intrinsic quality of the product in terms of the added value to the quality of milk and cheese deriving from the forage; |
| 5. animal welfare. | 5. specific culture of the people: identity, history, memory, traditions. |

These functions correspond therefore to provisioning and cultural ESs. Only the group of Predazzo evoked regulating (i.e., water cycle) and supporting (i.e., biodiversity) ESs. In sum, for both groups the economic and social dimensions of summer farm activities were the most important characteristics.

During the second session, the ranking of the ESs obtained from the online questionnaires (see Figure 5) was confirmed, except for the group of Primiero, which agreed that the ES "production of high-quality dairy products" should be the most important one. The focus groups confirmed also that all the ESs proposed were perceived positively by the local stakeholders. However, the participants agreed that the positive role of summer farms on the ESs varies according to the category of grazing livestock, e.g., it is important that cattle breeds and categories suitable for Alpine pastures are used, and to the adoption of appropriate pasture management practices, as a correct stocking rate.

In addition, the participants suggested that the ESs "improved soil carbon storage" and "contribution to mitigation of climate change" were too technical and difficult to understand, and therefore that it is necessary to strengthen the information on these ESs.

## 4. Discussion

This study demonstrated that local stakeholders and tourists have a different perception of the summer farms and of the associated private and public benefits, as indicated by the comparison of the semantic groups deriving from induced words. They also differed in their social valuation of a list of ESs, as indicated by the questionnaires and, for stakeholders, also by the focus groups. In addition, a lack of knowledge on supporting and regulating ESs emerged in both categories of actors. In the following, we will first discuss the differences in perception and social valuation between stakeholders and tourists, then address the gaps in knowledge, and finally conclude by commenting on the relevance of our results for the sustainability of summer farms and the associated ESs.

The semantic groups most frequent in the induced words of stakeholders indicated that their perception of summer farms was mainly focused on the provisioning ESs, through a knowledge of the role of farming practices and pasture management in the production process. In contrast, provisioning ESs were mostly ignored by tourists, who focused mainly on general benefits related to regulating and cultural ESs. In fact, tourists did not seem to associate summer farms with grazing livestock systems, as showed by the low frequency of the words "livestock" and "Alpine pastures". Moreover, Zuliani et al. [48] demonstrated a gap between the consumers' conception of mountain farming systems and the actual farming practices. The only semantic group that was highly frequent in both categories was "products". This group is clearly related to provisioning ESs, but may have also a cultural meaning because of the typicality and of the gastronomic value that can be attributed to the products of summer farms. We suggest that for tourists this latter dimension is prevalent: they visit summer farms to enjoy the scenery and taste products that they perceive as typical and unique. We also observed that the perception of summer farms by tourists did not depend on their personal attributes. On the contrary, Schirpke et al. [33] found that this perception in an Alpine landscape differed between social groups stratified by gender, age and origin. However, our sample of tourists was deliberately chosen to represent visitors of summer farms, and therefore might have been less heterogeneous.

The eight ESs proposed in association with summer farms were all assessed positively by both local stakeholders and tourists. Similarly, Faccioni et al. [31] showed that local stakeholders and the general population valued positively a range of ESs related to mountain livestock systems. Comparably, Lamarque et al. [49] found that the ESs linked to "high-quality products" and "maintenance of traditional landscapes" were among the most important ESs for both categories. Zoderer et al. [50] found that the landscape beauty was the most frequently perceived ES by tourists in South Tyrol. In this study, statistically significant differences between the assessment of tourists and local stakeholders emerged only for three ESs (Figure 5). Thus, we found that tourists valued more than local stakeholders the supporting ESs of biodiversity and the cultural dimension of high-quality products, and less that of the regulating ES of contrasting shrubs invasion on pastures. The provisioning of forage is central for the farmers and hence shrub encroachment is an important issue for them, especially because this process has recently increased

due to abandonment of Alpine grasslands [19]. Moreover, Bernués et al. [51] demonstrated that citizens gave more importance to cultural ESs, while farmers gave more importance to the regulating ESs that were directly connected with their activity (e.g., prevention of forest fire and soil fertility).

The positive assessment that local stakeholders declared for the ESs associated with the summer farms was not connected with a clear knowledge of the ESs concept, as was outlined in the online questionnaires (only 25.7% of respondents claimed to know the definition). Moreover, during the focus groups we often had to use the term "added value" to make the concept of ESs clearer to the participants. This fact was also noticed by Bernues et al. [51], who observed that in their focus groups no one knew the ESs concept. Moreover, Lamarque et al. [49] showed that the farmers interviewed for their study had never heard about the concept of ESs. More particularly, we observed an important lack of understanding of the ESs linked to soil carbon storage and climate change mitigation. This gap in knowledge seems to be general, because it was shared with the tourists. Interestingly, it seemed to influence the assessment of ESs by both actors, who tended to value these two ESs less than the others proposed. We argue that this is an important issue, especially because it contrasts with the importance of regulating ESs, and especially of those related with climate change, that is recognized within the scientific community. Indeed, Mengist et al. [28] outlined that in the papers that they reviewed the regulating ESs were those most frequently discussed (36.3%) among 317 different ESs, and particularly that carbon storage and climate regulation were those most frequently addressed. In contrast, provisioning ESs (27.1%), supporting ESs (19.9%) and cultural ESs (16.7%) were less frequently discussed. Martín-López et al. [29] found that a major part of the studies that they reviewed (28.2%) focused on regulating ESs. Martínez-Harms et al. [52] showed that carbon storage and carbon sequestration were addressed in 34% and 29% of 41 studies on ESs mapping conducted between 1995 and 2011.

The implications of our results for the sustainability of summer farms, and more generally of extensive livestock systems, can be various. Locally, but also in other Alpine regions, tourism promotion agencies coordinate, organize and promote events on summer farms for tourists to discover local dairy products or farming practices (e.g., "latte in Festa" "milk feast", "Albe in malga" "dawn in summer farm", "Desmontegada" "the return from summer farms"). Improving the communication between stakeholders and tourists might increase their mutual understanding of the links between summer farms, Alpine pastures and the associated ESs of mountain areas. Local stakeholders would benefit from an understanding of how tourists perceive and value supporting and cultural ESs, and tourists from an understanding of how the ESs that they value are linked with the farming practices. This would allow local stakeholders to appreciate the cultural values related to summer farms, and tourists to recognize the real features of mountain farming systems. Promoting informed and aware tourism experiences based on a shared knowledge of the ESs delivered by pasture-based livestock systems might benefit directly farmers through agritourism, and indirectly other local stakeholders, such as tourism and cultural operators, and in turn the economy of the rural communities, by increasing the tourism attractiveness and diversification.

In addition, the assessment of the perception and valuation of ESs by tourists, and more in general of the society, is useful to devise private actions aimed at increasing the economic sustainability of dairy food chains. In particular, the ESs provided by Alpine pastures, especially cultural ESs, might generate a higher market value of the products, as they are positively assessed by tourists. Interestingly, a study on the socio-economic valuation of ESs associated to extensive livestock systems in the province of Trento [30] found that a sample of residents locally and in bordering provinces declared a willingness to pay of 79.3€ for the preservation of water quality (regulating ES), 40.3€ for the conservation of biodiversity of grasslands (supporting ES), 35.5€ for the conservation of traditional agro-pastoral landscape (cultural ES), and 4.6€ for the provision of a larger number of high quality cheese types (provisioning ES). Therefore, the public valued the food chain more for the attributes of external quality of the products (the regulating, cultural and supporting ESs associated with the farming systems) than for the intrinsic quality of the cheese. This suggests that marketing strategies aiming to remunerate the

summer farm products based on an effective communication of the associated ESs might be successful. In addition, the collective actions needed to implement such strategies might be supported by the cooperative system that characterizes the Autonomous Province of Trento. This is especially relevant for our study area, where the cooperative dairies process summer farm milk into cheese and dairy products which are labeled as "summer farm products" to distinguish and valorize them as respect to the products obtained from milk produced in the lowland permanent farms.

Evaluating the social and economic values of ESs is also of paramount importance to implement public policies to effectively remunerate the mountain extensive livestock farming systems for the non-marketable benefits that they provide to the society [4,52]. While we did not address the economic valuation here, we assessed the level of knowledge and social valuation of ESs by local stakeholders and tourists, which is an essential preliminary step, also in light of the fact that, as shown by Bernués et al. [53], the perception of ESs may vary strongly across regions with the socio-cultural, economic and biophysical contexts.

The studies related to cultural ESs are increasing [5,15,50,54,55], although these ESs are difficult to measure and appreciations are still rare [33,54]. Uncovering the perceptions of cultural ESs by the different actors needs participatory approaches, as consultative methods (e.g., questionnaire) or group-based methods (focus groups) [51]. Furthermore, the economic valuation of the ESs, although subject to debate and criticism [56], is of growing interest to analyze the individuals' perception of cultural ESs [28]. The choice experiment or contingent valuation are for example common methods [2,30,31,57]. However, it is still difficult to translate the cultural ESs into monetary values and further research is needed.

Last, but certainly not least, our results stress the importance of filling the gap between the scientific understanding of ESs and the public awareness and knowledge of the ESs framework in general, and of specific supporting and regulating services in particular. The public, in our case both stakeholders and tourists, seem to be able to value the benefits that they somewhat perceive, even when unable to understand them within the ESs framework, but are unable to evaluate, or tend to value less, the less obvious ESs, which however might be very important. We concur therefore with Zuliani et al. [49] that strategies and policies should be implemented to foster communication on ESs and promote partnership between different stakeholders, in order to reduce the gap of knowledge between society and research. The development of innovative governance models, participatory approaches and integrated decision-support tools is strongly encouraged by the European Commission, in order to implement policy actions to support the sustainable development of rural areas. This paper focused on summer farms in a specific study area of the Eastern Alps, but the methodological approach could be extended to other contexts. The online questionnaire was used as a first step to have the individual opinion of relevant stakeholders. This kind of method risks being selective in terms of participation, as, for example, it is quite difficult to involve the farmers. The direct involvement of the Breeder Federation of Trento allowed the participation of individual farmers who have key roles in the study area (as coordinators of local farmers associations or of dairy cooperatives). The focus groups were then used to move from the individual opinions to the group opinion, and this is an example of a tool that can be used to favor multi-actors participatory approaches. Finally, the questionnaires proposed to the tourists in two very popular summer farms with agritourism activities allowed us to evaluate their perception (and knowledge) of summer farms and of the associated ecosystem services. The size of the sample of tourists interviewed is adequate to the specific aims of this study, but a larger survey could be proposed to the Autonomous Province of Trento to produce an extensive qualitative assessment of the public perception of summer farms ESs, which might be useful to support rural development policies and to define communication and marketing strategies [58]. The Alps are characterized by very heterogeneous farming systems/value chains, and the application of this multi-actors approach in different areas and at different scales could be used to identify specific opportunities and/or criticalities.

## 5. Conclusions

The present study showed that both local stakeholders and tourists have a generally positive perception of the ESs associated with summer farms and their products in an Alpine Italian area. However, we also found that both actors have important gaps in their knowledge of the ES concept and of crucial regulating and supporting ESs (i.e., carbon storage and mitigation of climate change). Secondly, local stakeholders perceive as being most important the provisioning or regulating services associated with the farming practice, while tourists perceive as being most important the regulating and cultural (recreational) ESs. This indicates that, in order to promote public and private actions to valorize the summer farms and their products on the basis of the associated ESs, it is crucial that these ESs are efficiently communicated, particularly those that are not recognized or little understood despite being very important. This would allow the tourists to acknowledge the real features of mountain farming systems, and, on the other hand, the local stakeholders to include a higher cultural dimension in their perception. In addition, scientific research should take more into consideration the perceptions of these actors who highly regard cultural and provisioning ESs. In fact, the concept of ESs shows a high potential to support the implementation of both private strategies and public policies to improve communication and increase knowledge on ESs, and to provide payments for specific ESs. These two approaches are both instrumental in ensuring the economic, but also the social and the environmental sustainability of summer farms and traditional mountain farming systems.

**Author Contributions:** Conceptualization, C.P. and E.S.; Methodology, C.P., R.D.R. and E.S.; Software, C.P.; Validation, M.R., S.B., D.G., and E.S.; Formal Analysis, C.P.; Investigation, C.P.; Writing—Original Draft Preparation, C.P.; Writing—Review & Editing, C.P., M.R., S.B., and E.S.; Supervision, E.S.; Project Administration, D.G.; Funding Acquisition, D.G., S.B., and E.S. All authors have read and agreed to the published version of the manuscript.

**Funding:** The research and the Grant of Carine Pachoud were funded by RDP 2014–2020, Autonomous Province of Trento, measure 16.1.1. and 16.1.2 (SMartAlp project, CUP C66D17000170008).

**Acknowledgments:** The authors thank the director and staff of the Paneveggio Pale di San Martino Natural park for their efforts in organizing the research. The authors thank the Breeder Federation of Trento and in particular Dott. Ilario Bazzoli for the support to the research.

**Conflicts of Interest:** The authors declare no conflict of interest.

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
