# Peer review of "Tourists and Local Stakeholders’ Perception of Ecosystem Services Provided by Summer Farms in the Eastern Italian Alps"

_sustainability, doi:10.3390/su12031095_

Round 1
Reviewer 1 Report
Specific comments on revisions have been inserted in the main file, as comments.

Author Response
Comments were embedded in the paper’s .pdf. We have accepted them all. Please see below replies:
Reviewer: comment at lines 26-27 submitted ms “I think this sentence should be reviewed. It is not clear”.
Author: we have revised the sentence, see lines 23-24 of revised ms
Reviewer: Comment at line 56 submitted ms “The structure of brackets, list elements and references of this sentence is correct, but a little bit confusing. Is it possible to make this sentence simpler?”
Author: we have revised the sentence, see lines 45-50 revised ms
Reviewer: Comment at line 89 submitted ms “I would check if represent is the correct verb for this sentence.”
Author: we have modified “represents” into “covers”, see line 91 revised ms
Reviewer 2 Report
General comments
Overall, the paper is interesting, tackling a subject of interest for the scientific community, different types of stakeholders involved in the dairy value chain from mountain areas, tourist and general public.
My main concerns refer to:
- the manner of presentation for the introductory aspects, especially regarding the present manner in which ecosystem services are valued through CAP measures and additional price differences that consumers might presently pay for ESs;
- the statistical relevance of samples selected with regard to the farms under analysis, tourists surveyed as well as local stakeholders interviewed and taking part in the two focus groups;
- similarly, a clarification should be achieved as to the manner in which the results obtained might contribute though particular measures to the support for traditional animal breeding activities in the mountain regions, to add to the current measures being applied.

Author Response
The paper analyzes the local stakeholders’ and tourists’ perception of the ecosystem services (ESs) provided by the traditional farming systems from the Italian Alps in the Province of Trento. The paper emphasizes the different types of Ecosystem Services, the differences in perception and social valuation between stakeholders and tourists, the implications that the results obtained have exerted on the sustainability of the traditional summer farming systems.
General comments
Reviewer: Overall, the paper is interesting, tackling a subject of interest for the scientific community, different types of stakeholders involved in the dairy value chain from mountain areas, tourist and general public.
My main concerns refer to:
- the manner of presentation for the introductory aspects, especially regarding the present manner in which ecosystem services are valued through CAP measures and additional price differences that consumers might presently pay for ESs;
Author: we have modified the introduction to briefly introduce how CAP measures and market policies could remunerate ESs, see lines 66-78 revised ms
Reviewer: the statistical relevance of samples selected with regard to the farms under analysis, tourists surveyed as well as local stakeholders interviewed and taking part in the two focus groups;
Author: please see answers to specific comments and similar comment of reviewer 3 (attached file)
Reviewer- similarly, a clarification should be achieved as to the manner in which the results obtained might contribute though particular measures to the support for traditional animal breeding activities in the mountain regions, to add to the current measures being applied.
Authors: we have enriched the discussion with comments on this topic, see lines 394-414 revised ms
I am suggesting some points the authors might like to consider in order to improve the manuscript.
Specific comments
Reviewer: Lines: 71-73: The issue brought forth, without questioning ESs associated to traditional farming systems, is how and how much consumers should pay for these extra non marketed ESs. Are there any studies tackling this issue?
Authors: we have implemented a brief discussion of this into introduction and mentioned relevant studies. See answers to general comments.
Reviewer: In the introductory part, authors could answer whether at present, with the current CAP and in the future, these ESs receive any support, such as the general or specific aid schemes that Italian or European farmers benefit from and whether these payments (or a part of them) may be aimed at ESs.
Authors: we have implemented a brief discussion of this into introduction and mentioned relevant studies. See answers to general comments.
Reviewer: Lines 74-75: The same type of approach as far as the possible higher market value of the local products is concerned.
Authors: we have implemented a brief discussion of this into introduction and mentioned relevant studies. See answers to general comments
Reviewer: Lines: 113-114: It is not entirely clear how representative the 2 summer farms are, as well as the 2 localities where the research was conducted. How many localities in the area have the same characteristics? What were the criteria applied to choose the two localities and the two farms?
Authors: we chose the two localities and the two summer farms as being representative of the provincial context, based on knowledge deriving from previous studies, and have modified the text to explain it and include citation of such studies. See lines 177-185 revised ms.
Reviewer: Lines 139-140: Why there was no Predazzo farmer selected in the focus group?
Authors: farmers were invited to participate to the online questionnaire and as reported in table 2 we have 4 participants. In the focus group of Predazzo the dairy cooperative representative is also a farmer, for this reason we considered acceptable the composition of the group (another farmer was invited but not able to participate for personal problem).
Reviewer: Line 159: To what extent are the 405 representative for the tourists in the region? What is the number of tourists in the region for the past few years? Can their representativeness be statistically proven in terms of number, as well as socio-demographic characteristics?
Authors: We did not aim at sampling all the tourists visiting the region, but only those visiting summer farms, which we have specified in the text (see lines 177-185, revised ms). For this we chose two representative summer farms and conducted interviews over a month period. The size of the sample of tourists interviewed is larger or similar to that of samples of other studies, which we have cited in the revised ms (see line 184 revised ms). We checked for effects of personal attributes of tourists on their perception, as stated in the submitted ms, and found no statistical differences.
Reviewer: Line 347-348: What kind of strategies should the farmer or local stakeholder use? Are the present ones insufficient?
Authors: we have modified the text to include a discussion on this point. See lines 394-432 revised ms
Lines 376-377 What kind of policies should be implemented to foster communication on ESs and promote partnership between different stockholders? Please expand the topic further.
Authors: we have modified the text to include a discussion on this point. See lines 394-432 revised ms

Reviewer 3 Report
The manuscript is interesting and current. The objective of the research is clear.
The methodology at present is rather weak and does not follow the usual patterns used in the analysis of stakeholders and social networks.
It is not clear whether the scheme used for the identification of subjects was the focus group, whether it followed the snow-ball sampling methodology or what other scheme of identification of individual stakeholders was followed.
Among the stakeholdeers there is a lack of institutions and other decision-making centres of primary importance.
Moreover, the categorisation of stakeholders in table 1, and as described in lines 132-134, deserves a discussion that guarantees the understanding of the classification criterion and the repeatability of the experiment.
Online questionnaires have many limitations related to their drafting and compilation that have not been adequately discussed.
The choice of statistical processing with a kruskal-wallis test denotes the weakness of the data, however the methodology section deserves a discussion of the choices made.
The results are rather weak and significant statistical differences between tourists and local stakeholders emerge in only three cases in Figure 5.
From the work it emerges that despite a good description of the perception of the investigated ES, the sample appears rather limited and should be strengthened and then repeated.
The discussion is interesting but a conclusive chapter is missing.
To deepen the methodological section I recommend some recent articles on similar areas of study:
https://www.mdpi.com/1999-4907/9/8/465
https://www.mdpi.com/1999-4907/9/8/463
https://www.mdpi.com/1999-4907/11/1/12
https://www.mdpi.com/1999-4907/9/8/468/xml?utm_source=TrendMD&utm_medium=cpc&utm_campaign=Forests_TrendMD_0
Author Response
The manuscript is interesting and current. The objective of the research is clear.
Reviewer: The methodology at present is rather weak and does not follow the usual patterns used in the analysis of stakeholders and social networks.
Authors: See answers below, where this comment is articulated into more detailed specific comments
Reviewer: It is not clear whether the scheme used for the identification of subjects was the focus group, whether it followed the snow-ball sampling methodology or what other scheme of identification of individual stakeholders was followed.
Authors: preliminary meetings were organised with the partners involved in the project to identify relevant stakeholders. The natural Park Paneveggio Pale San Martino proposed a first set of relevant stakeholders involved (directly or indirectly) in the dairy chains and summer farms management, and the initial list were implemented and classified according different categories. We have added a short paragraph (see lines 126-138 revised ms) to clarify and justify the scheme used for stakeholders identification
Reviewer: Among the stakeholders there is a lack of institutions and other decision-making centres of primary importance.
Authors: the identification of stakeholders considered the institutions relevant for the local management of summer farms and valorisation of products. In the Autonomous Province of Trento a central role is played by the Province and by local administrators, that have been involved in all the steps of the research. ,
Reviewer: Moreover, the categorisation of stakeholders in table 1, and as described in lines 132-134, deserves a discussion that guarantees the understanding of the classification criterion and the repeatability of the experiment.
Authors: We identified local stakeholders and did not want to classify them into categories of relevance. The groups are now indicated in material and methods (see lines 135-138 revised ms)
Reviewer: Online questionnaires have many limitations related to their drafting and compilation that have not been adequately discussed.
Authors: we have added a short discussion on such potential limits, see lines 139-143 revised ms
Reviewer: The choice of statistical processing with a kruskal-wallis test denotes the weakness of the data, however the methodology section deserves a discussion of the choices made.
Authors: We used this one-way test, as in other published papers which we have cited (citation 26 revised ms), because we wanted to compare how the two groups scored different ESs. We did not want, and think would not be correct, to compare the scores of the different ESs.
Reviewer: The results are rather weak and significant statistical differences between tourists and local stakeholders emerge in only three cases in Figure 5.
From the work it emerges that despite a good description of the perception of the investigated ES, the sample appears rather limited and should be strengthened and then repeated.
Authors: The fact that in some of the comparisons the differences between local stakeholders and tourists in fig 5 are not significant has been discussed (see lines 255-258 revised ms).
Reviewer: The discussion is interesting but a conclusive chapter is missing.
Authors: we have added a ”Conclusion” paragraph
Reviewer: To deepen the methodological section I recommend some recent articles on similar areas of study
https://www.mdpi.com/1999-4907/9/8/465
https://www.mdpi.com/1999-4907/9/8/463
https://www.mdpi.com/1999-4907/11/1/12
https://www.mdpi.com/1999-4907/9/8/468/xml?utm_source=TrendMD&utm_medium=cpc&utm_campaign=Forests_TrendMD_0
Author: thank you for the useful suggestions. The spatial and social contexts of the papers suggested (one of them is the result of a COST action) require indeed a methodological approach more complex and structured than that we used in this study. We have integrated in discussion the potential of further expanding our results with such approaches and cited the more pertinent of such papers.

Round 2
Reviewer 2 Report
I am pleased with the authors' response and the additions to their paper.
Author Response
Thanks for your positive comments
Reviewer 3 Report
The text in the current form appears to be improved, however, there are some elements to correct.
- The fonts used, especially in the new text inserted after revision, do not correspond to those in the journal, see L 139 and others.
- the digits of the currency in € use the decimal separator with the point "." and not the comma ",", see lines 392-394
- the readability of figures can be improved
- the text must be revised by an English native speaker
Author Response
The text in the current form appears to be improved, however, there are some elements to correct.
- The fonts used, especially in the new text inserted after revision, do not correspond to those in the journal, see L 139 and others.
AU: we have revised the manuscript according to the Template of the Journal
- the digits of the currency in € use the decimal separator with the point "." and not the comma ",", see lines 392-394
AU: thank you, we have corrected this mistake
- the readability of figures can be improved
AU: we have modified the figures to improve the readability
- the text must be revised by an English native speaker
AU: the manuscript was revised to improve the English